

# Diversity, spatial distribution and activity of fungi in freshwater ecosystems

Cécile Lepère[1], Isabelle Domaizon[2], Jean-Francois Humbert[3], Ludwig Jardillier[4], Mylène Hugoni[5] and Didier Debroas[1]

[1] Laboratoire: Microorganismes: Génome et Environnement, Université Clermont Auvergne, Clermont-Ferrand, France
[2] CARRTEL, Université Savoie Mont Blanc, INRA, Thonon Les Bains, France
[3] iEES Paris, Sorbonne Université, INRA, Paris, France
[4] Unité d'Ecologie, Systématique et Evolution, CNRS, Université Paris-Sud, AgroParisTech, Université Paris-Saclay, Orsay, France
[5] CNRS, UMR5557, Ecologie Microbienne, INRA, UMR1418, Université Lyon 1, Villeurbanne Cedex, France

## ABSTRACT

High-throughput sequencing has given new insights into aquatic fungal community ecology over the last 10 years. Based on 18S ribosomal RNA gene sequences publicly available, we investigated fungal richness and taxonomic composition among 25 lakes and four rivers. We used a single pipeline to process the reads from raw data to the taxonomic affiliation. In addition, we studied, for a subset of lakes, the active fraction of fungi through the 18S rRNA transcripts level. These results revealed a high diversity of fungi that can be captured by 18S rRNA primers. The most OTU-rich groups were Dikarya (47%), represented by putative filamentous fungi more diverse and abundant in freshwater habitats than previous studies have suggested, followed by Cryptomycota (17.6%) and Chytridiomycota (15.4%). The active fraction of the community showed the same dominant groups as those observed at the 18S rRNA genes level. On average 13.25% of the fungal OTUs were active. The small number of OTUs shared among aquatic ecosystems may result from the low abundances of those microorganisms and/or they constitute allochthonous fungi coming from other habitats (e.g., sediment or catchment areas). The richness estimates suggest that fungi have been overlooked and undersampled in freshwater ecosystems, especially rivers, though they play key roles in ecosystem functioning as saprophytes and parasites.

## INTRODUCTION

Molecular diversity of microbial eukaryotes in aquatic ecosystems is far less investigated than their prokaryotic counterparts. This is even more striking for particular groups such as fungi that attract very little interest. This bias partly results from their supposed low abundances (e.g., ∼1% of total marine eukaryotes (*Massana & Pedros-Alio, 2008*)) that suggests fungi have little ecological importance in aquatic ecosystems. However, rare organisms can play crucial roles in ecosystem functioning but more importantly recent studies have revealed much larger proportions of fungi than previously observed, as well as high taxonomic richness in different marine (*Le Calvez et al., 2009*; *Gao, Johnson &*

Corresponding author
Cécile Lepère, cecile.lepere@uca.fr

*Wang, 2010*; *Orsi, Biddle & Edgcomb, 2013*; *Richards et al., 2012*; *Lepère et al., 2015*) and freshwaters environments (*Monchy et al., 2011*; *Ishii, Ishida & Kagami, 2015*; *Duarte et al., 2015*; *Lepère et al., 2016*). The extent of fungal biodiversity is therefore likely underestimated (*Scheffers et al., 2012*) though diversity estimates based on molecular data suggest that it can range between 0.5 and 10 million species (*Hawksworth, 2001*; *O'Brien et al., 2005*; *Mora et al., 2011*; *Bass & Richards, 2011*; *Blackwell, 2011*).

Despite this putative rich biodiversity, functional roles of aquatic fungi, for which only 3,000–4,000 species have been recorded, remain poorly characterized (*Pautasso, 2013*; *Rambold, Stadler & Begerow, 2013*). They are mainly known as decomposers of leaves in rivers, mangroves and wetlands (*Seena, Wynberg & Bärlocher, 2008*; *Gulis, Kuehn & Suberkropp, 2009*) and as parasites of phytoplankton and zooplankton in lake ecosystems (*Jobard, Rasconi & Sime-Ngando, 2010*). A decade ago, fungi were divided into four main phyla: Basidiomycota, Ascomycota, Zygomycota and Chytridiomycota (*James et al., 2006*). However, phylogeny of fungi is still unresolved and several phyla, classes and orders of basal fungi have been determined, since. For example, *Corradi (2015)* highlighted that Cryptomycota forms a new phylum in which we can find Microsporidia, Aphelids and Rozellids. The vast majority of this phylum is characterized by environmental sequences and gathered under the term ''dark matter fungi'' (*Grossart et al., 2015*). These fungi are mostly zoosporic and ''old'' in term of evolution since they diverged from the remaining fungi 710–1,060 million years ago (*Lücking et al., 2009*). These basal fungi are distant from cultured and described fungi. Aquatic environments are thus likely to host a high number of uncharacterized groups (*Grossart & Rojas-Jimenez, 2016*).

With the goal to draw up an inventory of existing knowledge on the diversity, distribution and ecology of aquatic fungi, we (1) combined all publicly available fungal 18S ribosomal RNA gene sequences produced by high-throughput sequencing approach from freshwater studies (2) compared fungal composition across environments (3) analysed a new set of data presenting 18S rRNA transcripts abundance on a sub-sample of 8 lakes in order to characterize the active part of the community.

## MATERIALS AND METHODS

### Data collection of 18S rRNA genes in public databases

In this work, we collected a set of publically available data that were related to HTS (pyrosequencing and Illumina with MiSeq technology) of the V4 region of the gene encoding for 18S rRNA (Tables 1, 2 and *Debroas et al. (2017)*). These sequences were obtained from freshwater ecosystems (25 lakes and ponds, and four rivers), sampled at various depths and dates (long term or periodically), and/or obtained from various size fractions (Table 1).

In this analysis, we included external references such as V4 amplicons sequenced from a few non-freshwater ecosystems (marine ecosystems and environments characterized by salinity gradients) to compare environments and define spurious OTUs (i.e., singletons, see below) (Table 1).

**Table 1** High throughput sequencing (HTS) data used in this analysis.

| Ecosystems | References | Geographic area | Size fraction | Primers | HTS |
|---|---|---|---|---|---|
| **Freshwater environments used for taxonomic affiliation** | | | | | |
| **Lakes** | | | | | |
| Pavin | *Debroas, Hugoni & Domaizon (2015)* | Massif central (France) | 0.2–5 μm | NSF573-NSR951 | 454 |
| Bourget | *Debroas, Hugoni & Domaizon (2015)* | Alps (France) | 0.2–5 μm | NSF573-NSR951 | 454 |
| Leman | *Mangot et al. (2013)* | Alps (France) | 0.2–5 μm | NSF573-NSR951 | 454 |
| LakeA | *Charvet et al. (2012)* | Arctic | 0.2–3μm | E572F-E1009R | 454 |
| LakeWH | *Charvet, Vincent & Lovejoy (2014)* | Arctic | | E572F-E1009R | 454 |
| Aydat, Anterne, Godivelle, Pavin, Bourget, Sep, Villerest | *Lepère et al. (2013)* and *Taib et al. (2013)* | Massif central (France) and Alps | 0.2–5 μm | NSF573-NSR951 | 454 |
| LaClaye EtangVallees SaintRobert Garbard | *Simon et al. (2014)* and *Simon et al. (2015)* | Chevreuse Valley (France) | 0.2–5 μm | EK-565F18s-EUK -1134-R- UNonMet | 454 |
| FAS3 FAS4 | *Kammerlander et al. (2015)* | Alps | >0.65 μm | TAReukV4F-TAReukREV3 | 454 |
| HL5 HL15 | | Himalaya | >0.65 μm | | 454 |
| A iguebelette | MG-RAST:4703954.3 to 4703966.3 | Alps (France) | <50 μm | NSF573-NSR951 | 454 |
| Leman | MG-RAST: 4703954.3 to 4703966.3 | Alps (France) | 0.2–50 μm | NSF573-NSR951 | 454 |
| Vichy, Aydat, Eguzon, Pavin, Fades, Cournon, Grangent, RocheMoines | MG-RAST: 4703954.3 to 4703966.3 | Massif central (France) | 0.2–50 μm | 515F-951R | MiSEQ |
| **Rivers and brooks** | | | | | |
| RiverA and RiverB | *Bricheux et al. (2013)* | France | Biofilm | 528FE-1193E | 454 |
| SaintAnneBrook | *Simon et al. (2014)* and *Simon et al. (2015)* | Chevreuse Valley (France) | 0.2–5 μm | 565F-1134R | 454 |
| Morcille | MG-RAST: 4703954.3 to 4703966.3 | France | Biofilm | NSF573-NSR951 | 454 |
| **External references: saline environments and/or DNA amplifying by specific primers** | | | | | |
| Ngallou : hypersaline ponds | Roux et al. (2016) | | | NSF573-NSR951 | 454 |
| Blanes Naples Oslo Roscoff Varna | *Logares et al. (2009)* | | | V4 | 454 |
| Saline lakes | *Wang et al. (2010)* | | | A-528F B-1055R | 454 |
| Arctic Ocean | *Killias et al. (2014)* | | | 528F-1055R | 454 |
| Coastal | *Bachy et al. (2014)* | | | Ciliates Tin454-18SFw - Tin454-18SRev | 454 |

**Table 2  Richness of main taxonomic groups of fungi in freshwater ecosystem.**

|  | S. obs | S. chao1 | se. chao1 | S. ACE | se. ACE |
|---|---|---|---|---|---|
| **Freshwater** |  |  |  |  |  |
| Fungi | 25,771 | 25,713 | 61 | 25,741 | 69 |
| Ascomycota | 3,339 | 3,339 | 1 | 3,350 | 26 |
| Basidiomycota | 4,061 | 4,061 | <1 | 4,063 | 27 |
| Cryptomycota | 9,559 | 9,559 | <1 | 9,562 | 41 |
| Chytridiomycota | 3,927 | 3,927 | <1 | 3,932 | 28 |
| **Lakes** |  |  |  |  |  |
| Fungi | 17,026 | 17,419 | 27 | 18,057 | 62 |
| Ascomycota | 2,920 | 2,962 | 8 | 3,044 | 25 |
| Basidiomycota | 2,142 | 2,153 | 4 | 2,188 | 21 |
| Cryptomycota | 4,723 | 5,229 | 40 | 5,619 | 37 |
| Chytridiomycota | 3,002 | 3,035 | 7 | 3,122 | 26 |
| **Rivers** |  |  |  |  |  |
| Fungi | 12,453 | 12,757 | 23 | 13,279 | 48 |
| Ascomycota | 1,009 | 1,191 | 26 | 1,317 | 20 |
| Basidiomycota | 2,212 | 2,228 | 5 | 2,282 | 19 |
| Cryptomycota | 6,891 | 7,005 | 14 | 7,219 | 36 |
| Chytridiomycota | 1,323 | 1,360 | 8 | 1,433 | 16 |

### RNA extraction and 18S rRNA amplification

To go further than the analysis of 18S rRNA genes we sampled eight lakes which are included in the meta-analysis (Vichy, Aydat, Cournon, Grangent, Roche aux Moines, Eguzon, Fades and Pavin, Table 1) to studied the 18S rRNA transcript level. They included both natural and human-made waterbodies with considerable heterogeneity in terms of surface area and catchment characteristics. Sampling was carried out during the homothermy period in winter 2013–2014 as described in *Hugoni et al. (2015)*. Water samples were integrated in the photic zone of each lake over the deepest part of the lake. A sub-sample of water (300 mL) was pre-filtered through 150 and 50-μm pore-size filters and collected on 0.2-μm pore-size (pressure < 10 kPa) polycarbonate filters (Millipore) before storage at −80 °C until nucleic acid extraction.

The nucleic acids extraction method was done as described in *Hugoni et al. (2015)*. Briefly, after thermic and enzymatic cell lysis, the AllPrep DNA/RNA kit (Qiagen, Valencia, CA) was used. RNA samples were tested for the presence of contaminating genomic DNA using PCR and then reverse transcribed with random primers using SuperScript® VILO (Invitrogen). Amplification of the V4 region of the 18S rRNA genes among cDNA was performed using the universal primer 515F (GTG-YCA-GCM-GCC-GCG-GTA, (*Caporaso et al.,2010*) and the eukaryotic primer 951R (TTG-GYR-AAT-GCT-TTC-GC). Sequencing was achieved by the Genoscreen platform (Lille, France), using an Illumina Miseq paired-end chemistry.

## Sequence analysis procedures

All sequence data (public databases and new data) were examined against the following quality criteria: For the pyrosequencing data: (i) no Ns in the nucleotide sequence, (ii) quality score $\geq$ 23 according to the PANGEA process (*Giongo et al., 2010*), (iii) a minimum sequence length of 200 bp, and (iv) no sequencing errors in the forward primer. The MiSEQ data were assembled with the USEARCH tool (usearch v7.0.1090_i86linux64) (*Edgar, 2013*) and examined in relation to the previous criteria as well as for the absence of errors in the reverse primer. Putative chimeras and homopolymers were detected by UCHIME (*Edgar et al., 2011*) and a customised script (https://github.com/panammeb/stable/blob/master/modules/check_homopolymers.pm).

The clean freshwater reads were clustered at a 95% similarity threshold (*Mangot et al., 2013*; *Lepère et al., 2016*; *Debroas et al., 2017*) with USEARCH 7.0 (option: cluster_fast) (*Edgar, 2013*) to identify representative OTUs. Clean data for the external references (e.g., sequences from organisms in marine environments) and selected sequences from the SILVA database named RefEUKs (*Debroas et al., 2017*) were mapped on the representative OTUs to define them. This procedure allowed us to remove the singletons. A singleton in freshwater environments was therefore defined as a read sequenced only once, regardless of the environment, and that was absent in the SILVA database.

## Taxonomic affiliations

The representative OTUs were affiliated by similarity and phylogeny with a curated reference sequences named RefEUKs (https://github.com/panammeb/). These eukaryote references were extracted from the SSURef SILVA database (*Pruesse et al., 2007*) according to the following criteria: length >1,200 bp, alignment quality score >75%, and a pintail value >50. In addition, the taxonomy of this reference database was modified to include typical freshwater lineages (e.g., fungi) defined in previous studies (e.g., (*Debroas, Hugoni & Domaizon, 2015*). After a comparison of the OTUs with the RefEUKs by a similarity approach (USEARCH tool), trees of OTUs with their closest reference sequences were built in FastTree (*Price, Dehal & Arkin, 2010*) (see *Debroas et al., 2017*) for detailed pipeline). Taxonomic assignment was conducted according to two methods: nearest neighbour (NN), and last common ancestor (LCA) affiliations (*Liu et al., 2008*).

## Comparing representative OTUs with reference sequences from a public database

To compare freshwater OTUs to reference 18S rRNA gene sequences found in the public database, we used two criteria: similarity and phylogenetic metrics. In the first approach, OTUs were compared to the SSURef SILVA database and were restricted to the total or cultivated eukaryotes using BLAST. In the second, different phylogenetic indices (*Swenson, 2009*) were computed from the trees generated in the pipeline described above. The "X depth/deeper" is defined as the average distance to the deepest node in the tree (*Pommier et al., 2009*). These various indices were computed using R software with the packages "picante" (*Kembel et al., 2010*), "Geiger," and "ape" (*Paradis, Claude & Strimmer, 2004*), and were implemented in PANAM (*Taib et al., 2013*).

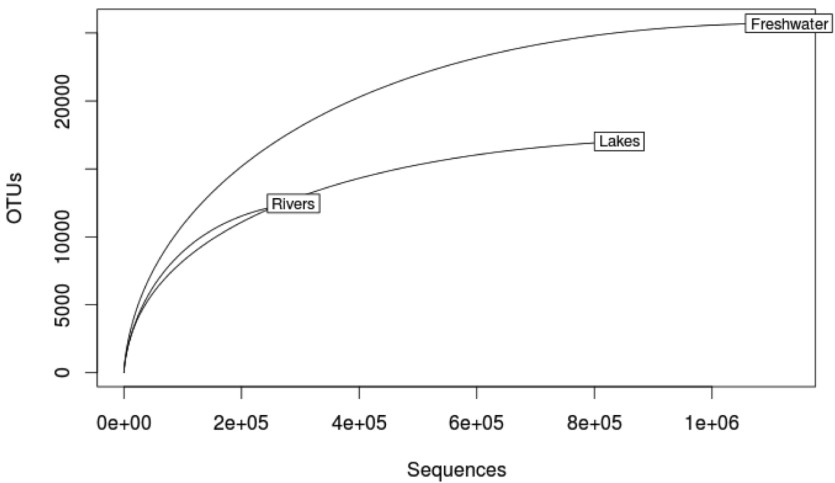

**Figure 1** Rarefaction curves for lakes, rivers and freshwater ecosystems (rivers + lakes) computed from fungal OTUs.

## Statistics

Different estimators were used to infer the taxa richness of the planktonic eukaryotes: non-parametric estimators (Chao1, ACE) and indices based on the rank-abundance curves. These estimators were computed with Vegan (*Dixon, 2003*) and the SPECIES packages (*Wang, 2011*) implemented in R.

### TSA-FISH (Tyramide Signal Amplification-Fluorescent In situ Hybridization)

On two lakes (Aydat, Pavin) included in the meta-analysis we performed TSA-FISH to detect Cryptomycota. The probe used was the LKM11-01 (*Mangot et al., 2009*) and the protocole is described in *Lepère et al. (2016)*.

## RESULTS

### Richness and community composition of freshwater fungi

The rarefaction curves built from OTUs show that a plateau is reached for lakes and for freshwater environments considered as a whole (lakes + rivers). On average 37,680 reads per samples were obtained. No plateau is obtained for river ecosystems (Fig. 1). From 1.6 million of reads, our analysis recovered 19,008 fungal OTUs at a similarity threshold of 95% (*Debroas et al., 2017*). The estimated OTU richness in rivers, lakes and freshwater ecosystems (lakes + rivers) vary according to the estimators (Table 2). The majority of the fungi retrieved in freshwater are represented by Dikarya (Ascomycota and Basidiomycota), Cryptomycota and Chytridiomycota. All estimators suggest that Cryptomycota is the richest group in both lakes and rivers while the lowest richness is found for the Basidiomycota in lakes and Ascomycota in rivers (Table 2).

When looking at the OTU taxonomic level in freshwater (lakes + rivers), 47% of fungal OTUs (26% of reads) are affiliated to Dikarya (1/3 to Basidiomycota, 2/3 to Ascomycota) followed by basal fungi: Cryptomycota (17.6% OTUs; 15% reads) and
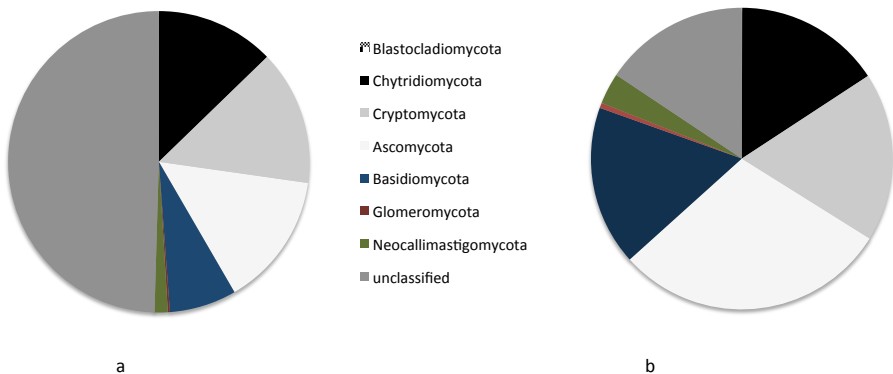

**Figure 2** Taxonomic identities of fungal OTUs (A) and reads (B) in freshwaters.

Chytridiomycota (15.4% OTUs; 12% reads) (Fig. 2). Moreover, 16% of the OTUs can't be affiliated and are grouped under "unclassified fungi". These "unclassified fungi" represent 50% of the reads (Fig. 2). Most of the OTUs with a taxonomic affiliation belonged to the subkingdom Dikarya (Ascomycota, Basidiomycota) in both lakes and rivers. The majority (almost 85%) of Ascomycota OTUs group within the *Pezizomycotina* (Fig. S1) and the most represented class within this subphylum is the *Dothideomycetes*. The rest of the Ascomycota OTUs belong to Ascomycota yeasts (subphylum: *Saccharomycotina* subphylum, class: Saccharomycetes).

A high proportion of Basidiomycota sequences are also found in our dataset, with the majority being yeast fungi. The three major clades of Basidiomycota are represented: *Agaricomycotina, Pucciniomycotina* and *Ustilaginomycotina* with the *Agaricomycotina* accounting for 53% of the Basidiomycota OTUs (Fig. S1).

## Spatial distribution of freshwater fungi

All phyla were detected in lakes and rivers but several are found at very low proportion, less than 1% of the OTUs and less than 2% of the reads (i.e., Blastocladiomycota, Glomeromycota). At a finer taxonomic resolution (i.e., OTUs level), 3,369 OTUs (18%) are found in both lakes and rivers while 12,424 (66%) are restricted to lakes and 3,215 to rivers (16%) (Fig. S2). Regarding the basal fungi, Cryptomycota OTUs predominate over Chytridiomycota in river while OTUs belonging to Chytridiomycota are more abundant in lakes as well as the proportion of Dikarya (Fig. S3). When considering only lacustrine ecosystems, the number of OTUs shared by the different lakes decrease exponentially with the number of lakes leading to the fact that 61% of the OTUs (9642) are restricted to one lake only. No OTU is shared by more than 19 lakes (over the 25 considered) and only 0.2% of the OTUs are shared by more than twelve lakes. There is a link between the most ubiquitous taxa and their abundances (i.e., number of reads), with the most widely distributed falling within the most abundant OTUs (Fig. 3). The top 10 most abundant OTUs (representing 7,334 reads per lake on average) are shared by a minimum of eight lakes (Fig. 3). However, there is no linear relationship between abundance and distribution. For example, one Cryptomycota OTU (Leman_S2331067) belongs to the 32 more abundant OTUs though it
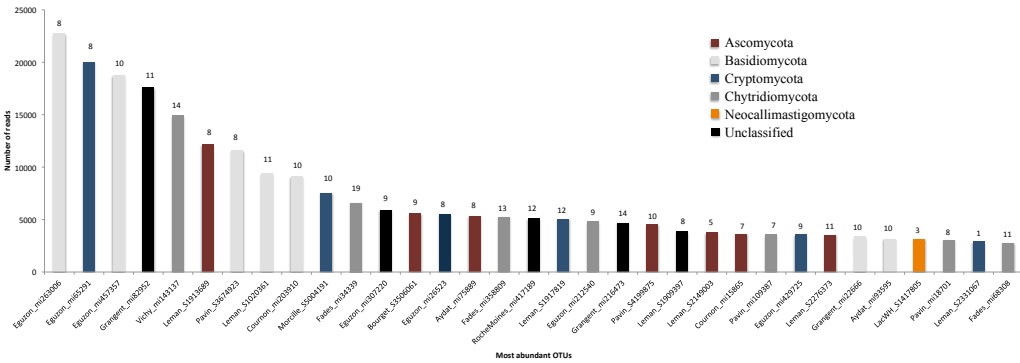

**Figure 3  Abundance (Number of reads) of top 32 fungal OTUs.** The identity number of the respective OTU is shown below the bars. Colours represent the fungal phyla and the numbers at the top of the bars represent the numbers of lakes where the OTU can be found.

is detected in one lake only. The most shared OTUs between lakes belong to Chytrids and Dikarya while the more abundant belong to Basidiomycota and Cryptomycota (Fig. 3).

789 fungal OTUs detected in freshwater are also found in marine environments displaying different salinities (Table S1). The majority of these OTUs (582) are detected in lakes. Most of the OTUs (224) shared by marine environments and lakes belong to Ascomycota and Chytrids (145). Rivers and marine environments share a majority of Ascomycota (82) and Cryptomycota (44) OTUs (Fig. S2).

## Active fungi in freshwater lakes

By targeting the 18S rRNA transcripts on a reduced number of lakes found in this meta-analysis (Vichy, Aydat, Cournon, Grangent, Roche Moines, Eguzon, Fades and Pavin, Table 1), we found that 30% of the eukaryotes OTUs are represented by fungi while they represent only 7.2% of the reads. 18S rRNA transcript sequencing showed the same dominant phylogenetic groups assessed by 18S rRNA genes analysis: Dikarya, Cryptomycota, Chytridiomycota as well as a large proportion of unclassified sequences. The mean rRNA transcripts: rRNA genes ratio computed from each OTU is 0.82. On average 13.25% of the fungal OTUs were active (Fig. 4A) (i.e., rRNA:rDNA ratio > 1). If we focus on the Dikarya which includes a broad diversity, some subphyla were more active than the others. Within the Basidiomycota, the Ustilaginomycotina did not seem to be active (Fig. 4B) whereas 28% of the Agaricomycotina were active (Fig. 4B). The most active subphylum within the Ascomycota is the Pezizomycotina (Fig. 4B) and the number of sequences in the 18S rRNA transcripts dataset was more abundant than in the 18S rRNA genes dataset also for Saccharomycotina and Mitosporic Ascomycota while the environmental sequences showed the opposite.

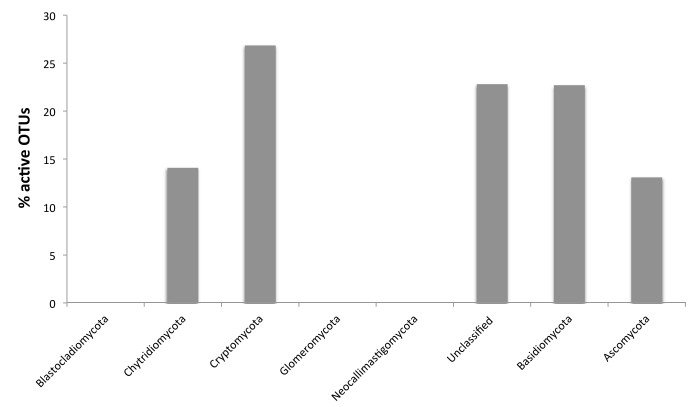
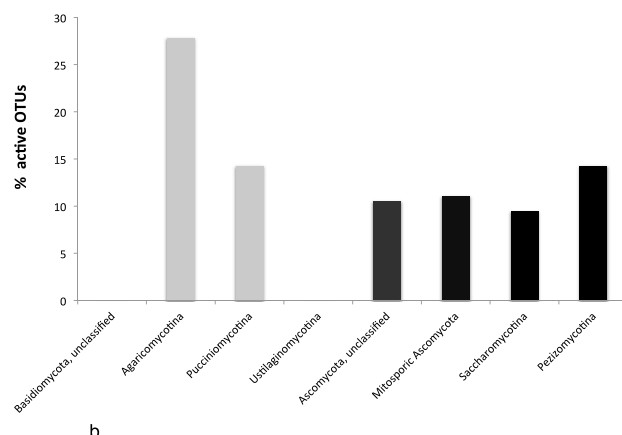

a                                                           b

**Figure 4** **% of active fungal OTUs (i.e., DNA OTUs found in the RNA dataset) (A), % of active Basidiomycota (grey) and Ascomycota (black) OTUs (B).**

# DISCUSSION

## 18S rRNA gene sequences

Until recently, fungi were one of the most understudied microbial groups in aquatic ecosystems, being more studied in terrestrial habitats. This trend has changed over the last decade and there is now more interest in deciphering the diversity and role played by fungal communities in aquatic ecosystems. Aquatic fungi have been studied through environmental clone library approaches (*Lai et al., 2007*) or high-throughput sequencing (HTS) using specific fungal primers (*Monchy et al., 2011*; *Arfi et al., 2012*; *Wang et al., 2014a*; *Wang et al., 2014b*; *Duarte et al., 2015*; *Zhang et al., 2016*). However, aquatic fungi are most often not considered on their own but rather studied as part of whole microbial eukaryotic community investigations (*Livermore & Mattes, 2013*; *Panzer et al., 2015*; *Hassett & Gradinger, 2016*; *Tisthammer, Cobian & Amend, 2016*). Therefore, the majority of the data available on aquatic fungi is 18S rRNA gene sequences. Recently, *De Filippis et al. (2017)* showed that 18S rRNA gene allow a more reliable quantification of fungi than the ITS region. ITS, which is the most accurate phylogenetic marker for fungi based on *in silico* simulations, promotes preferential amplification of shorter sequences and therefore leads to a biased view of taxa relative abundance. In freshwaters, the main phylogenetic markers used are the V4 (Table 1) and V9 regions (*Korajkic et al., 2015*) of the 18S rRNA and, more rarely, the V3 zone (*Nolte et al., 2010*). For an easier and more accurate taxonomic identification, we choose to focus on the V4 sequences since this is the largest dataset but also because this variable zone is present in almost all Sanger sequences deposited in GenBank. The sequences analyzed in this study have been obtained by the use of five "universal" primer sets which could introduce biases in the meta-analysis. Indeed, even if these primer sets target a similar V4 region of the 18S rDNA, they may lead to slightly different taxonomic compositions (*Debroas et al., 2017*). Moreover the different DNA extraction methods used in the studies considered in this analysis may produce bias. Indeed, the accuracy of the data depends on how well the DNA is extracted from

the environment, so that it accurately reflects the composition of the actual microbial community (*Djurhuus et al., 2017*). It is therefore difficult to disentangle the bias due to the primers and extraction methods from the effects of biogeography or environmental parameters. However, the use of several primer sets could be also an advantage taking into account that the use of different primers/markers leads to a better image of the diversity (*Shi et al., 2011*; *Debroas et al., 2017*). It's noteworthy that we chose to use the general term fungi even though the phylogeny of some groups cited in this paper is not yet fully elucidated.

## Taxonomic resolution

The mean size (∼400 bp) of the V4 amplicons gives taxonomy at a relatively broad resolution. For example, more than 90% of the Chytrids OTUs were identified at the genus level while 99% of the Cryptomycota OTUs are not identified below the phylum level. Altogether, a total of 52 genera of fungi were identified within nine phyla (including *Incertae sedis*). OTUs detected in one ecosystem only, were, for most of them, "unclassified fungi". When considering freshwaters (lakes + rivers), the sequence similarity of Cryptomycota OTUs was generally <80% to that of sequences available in the SILVA database whereas for Chytrids OTUs, this sequence similarity was on average 87%. Except for the Chytrids, fungal sequences found in rivers were characterized by lower similarities than the ones found in lakes suggesting for example that Dikarya are less known in rivers.

## Fungal communities in freshwater habitats

Meta-analyses are powerful analytical tools to decipher the structure and ecology of microbial communities (*ArchMiller et al., 2015*). When analyzing the global microbial eukaryotic diversity in freshwaters, using the same dataset as in our study, *Debroas et al. (2017)* showed that fungi represented 17% of this diversity. Freshwater fungi, analyzed in this paper, were affiliated to nine phyla, revealing the great richness that can be captured by the use of different universal 18S primers. Considering OTUs, Ascomycota and Basidiomycota were the most represented phyla in this dataset. The dominance of Dikarya is not usual in freshwater lakes. Indeed, the general assumption is that basal fungi (especially Chytrids) dominate the fungal community composition (*Monchy et al., 2011*; *Panzer et al., 2015*; *Comeau et al., 2016*; *Rojas-Jimenez et al., 2017*) while Dikarya tend to be dominant in marine ecosystems (*Panzer et al., 2015*). It's worth mentioning that Dikarya represented 26% of the fungal reads and present the lowest richness in freshwater. However, the low observed richness could be due to the marker used here which is relatively less resolving for this group than others we've surveyed (*Schoch et al., 2012*). The great majority of the Ascomycota OTUs fall indeed within the Pezizomycotina which include 32,000 species (*Kirk et al., 2001*). Pezizomycotina contains filamentous species that are ecologically diverse and the most represented class is the Dothideomycetes, which can have terrestrial and aquatic members (*Shearer et al., 2009*). Terrestrial filamentous fungi (e.g., associated or not with plants) can be introduced into lakes through spores and pieces of mycelia during inflowing stream, rainwater and wind events (*Voronin, 2014*). Most of the time these fungi fail to establish stable population in aquatic environments (*Graupner et*

*al., 2017*). Consequently, it is difficult to know if these fungi are truly aquatic (*Wurzbacher, Bärlocher & Grossart, 2010*). However, the rRNA analysis (eight lakes dataset) showed that these fungi were active in freshwater lakes. Filamentous fungi could also be associated with roots of aquatic macrophytes (*De Marins, Carrenho & Thomaz, 2009*; *Beck-Nielsen & Madsen, 2001*), similarly to fungi associated with roots of terrestrial plants. However, there is little knowledge about fungi in submerged roots. Pleosporales were also well represented. Species in this order inhabit various ecosystems, and are known as saprobes that decay plant material in freshwater (*Shearer et al., 2009*) and marine habitats (*Suetrong et al., 2009*). The rest of the Ascomycota OTUs belongs to Ascomycota yeasts. These yeasts live as saprobes, often in association with plants and animals. They are well distributed in deep-sea regions (*Bass et al., 2007*; *Nagahama et al., 2011*) and oxygen-depleted ecosystems (*Takishita et al., 2007*). Moreover, numerous Ascomycota are known to be pathogens of algae in marine systems (*Kohlmeyer & Kohlmeyer, 1979*; *Kubanek et al., 2003*).

Interestingly, a great number of Basidiomycota sequences were detected in our dataset. Even though recent molecular data suggest that the Basidiomycota diversity might be high in aquatic ecosystems (*Richards et al., 2012*; *Panzer et al., 2015*), less than 100 described species were isolated from aquatic sources (*Shearer et al., 2007*; *Jones & Fell, 2012*). Some OTUs were affiliated to the three major clades of Basidiomycota: Agaricomycotina, which represented more than half of the Basidiomycota OTUs and includes the vast majority of edible mushrooms forming spores and Pucciniomycotina as well as Ustilaginomycotina, which are known as plant parasites and sometimes have been found in association with aquatic invertebrates. For example, several OTUs are affiliated with the basidiomycete yeast *Rhodotorula* (Pucciniomycotina). It has been found in deep-sea tubeworms and bivalves as well as in different environmental DNA surveys (*Nagahama et al., 2003*). The rRNA sequencing surprisingly showed that the Agaricomycotina were the most active group within the Basidiomyota while the Ustilaginomycotina did not show any active OTUs. Even though Dikarya are identified as inhabiting terrestrial environments they were found active in freshwater lakes. They can therefore be directly involved in the trophic food web functioning. There could also be a continuous flow of active terrestrial Dikarya to the aquatic ecosystems through flooding for example (*Röhl et al., 2017*). Another explanation could be that these organisms were not active despite their high rRNA content. Chytridiomycota and Cryptomycota were also well represented in this freshwater dataset. Together they represent almost a third of the fungal OTUs/reads. These two phyla are included in the so-called DMF (Dark Matter Fungi), which encompass uncultured taxa belonging to early diverging branches of the fungal tree (*Grossart et al., 2015*). Chytridiomycota are well documented in freshwater lakes where they play various roles as saprobes and parasites (mainly of phytoplankton) (*Panzer et al., 2015*). Chytridiomycota zoospores are also a food resource for zooplankton (mycoloop) (*Gleason et al., 2008*). Chytridiomycota seemed to be relatively less active than Cryptomycota and Dikarya, only 8.4% of their OTUs were found active. Because of their undersampling, little is known about Chytridiomycota in rivers. Nevertheless, a recent study showed high abundances of Chytridiomycota zoospores in the Columbia river (*Maier & Peterson, 2016*).

In contrast to Chytridiomycota, Cryptomycota were discovered more recently in aquatic environments (*Jones et al., 2011*). Cryptomycota have been reported to account for only 0.02–4.5% of the total 18S rDNA sequences found in aquatic ecosystems (*Livermore & Mattes, 2013*). This group is highly-diverse (15 clades have been identified (*Lazarus & James, 2015*)), but it is almost exclusively known through environmental sequences. These fungi are found in a large range of ecosystems (*Jones et al. (2011)*; *Livermore & Mattes (2013)*, *Lazarus & James (2015))* without any specific clades to freshwater, soil, or marine systems (*Livermore & Mattes, 2013*). In freshwater a few investigations showed that they could act as parasites of phytoplankton (*Jones et al., 2011*; *Ishida et al., 2015*). Using fluorescent *in situ* hybridization, our investigations showed, indeed, several associations between diatoms (*Asterionella*) and Cryptomycota (Fig. 5) in a freshwater lake. All estimators suggest that Cryptomycota is the richest group in both lakes and rivers. In term of rDNA OTUs abundance, Cryptomycota dominate over Chytridiomycota in rivers. Moreover, Cryptomycota OTUs were usually less than 80% similar to the sequences deposited in databases in both lake and rivers and can't be identified at a high taxonomic resolution. Moreover, 17% of the Cryptomycota OTUs were active in Freshwater lakes and represented a large proportion of the total rRNA reads. This shows the need to enhance reference databases by increasing the sampling effort in freshwater ecosystems, especially streams.

## Fungal distribution across ecosystems

The low number of shared fungal OTUs among the diverse studies considered here suggests a high diversity and a low proportion of generalist fungi. Only a few fungal lineages were assigned at the species or genus level suggesting that even the more common lineages are poorly known or are missing from databases. 61% of the OTUs were found in one lake only. Interestingly, most of these OTUs have very weak supported affiliation (i.e., environmental samples). We cannot exclude the possibility that increasing further the sampling depth may lead to identify a higher number of OTUs characterized by a broader distribution. *Simon et al. (2015)* highlighted the importance of temporal surveys in the study of the microbial diversity. Such approach helps at detecting taxa that can occur at low frequency.

Our analysis showed that a very small number of fungal OTUs (5.5%) is shared between marine and freshwater ecosystems while up to 23% of the OTUs were shared between lakes and rivers. This confirms the data of *Panzer et al. (2015)* showing that freshwater fungal community structure differed significantly from all other habitats and of *Logares et al. (2009)* showing that fungi usually group into distinct marine and freshwater phylogenetic clusters. Molecular results from SSU rRNA and ITS1 region analyses also support the idea of a transition in fungal community structure along a salinity gradient (*Burgaud et al., 2013*). Until recently, marine fungi showed low diversity and abundance especially in the photic zone. However HTS has suggested that marine ecosystems contain more fungal diversity than previously thought (*Richards et al., 2015*). For example, *Livermore & Mattes (2013)* found evidence of considerable Cryptomycota diversity at the marine surface and *Richards et al. (2015)* showed that marine fungi include a large number of chytrid that had not been described before.

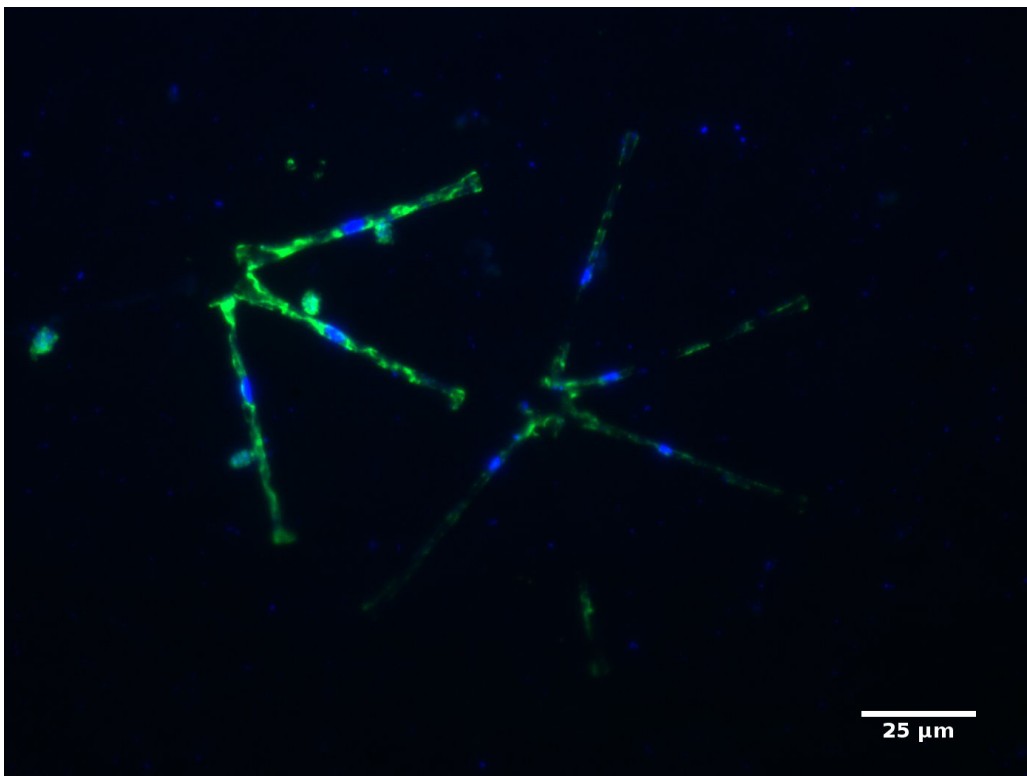

**Figure 5** Micrographs showing Cryptomycota cells (targeted with the LKM11-01 probe, Mangot et al. 2009) attached to the diatom *Asterionella* (21/05/2015 Aydat lake, France).

## CONCLUSION

This study represents the first in-depth inventory of published 18S rRNA gene sequence data on freshwater fungi and also adds some new data regarding potential activity of fungi in freshwater lakes. However, our knowledge on this phylogenetic group remains very limited. This is partly due to methodological limitations such as inaccurate methods for fungal morphological identification, non-specific molecular markers and to the lack of well-represented reference databases. We are indeed currently limited by the low representation of functional genes in the databases as well as the lack of sequenced fungal genomes, which prevent the exploration of the metabolic capacities of aquatic fungi through metagenomic and metatranscriptomic studies for instance, even though ongoing projects will likely reverse this bias such as the 1000 Fungal Genomes Project (*Grigoriev et al., 2014*). Our study highlights the need to increase the sampling effort at a global scale by conducting surveys in the most diverse set of aquatic ecosystems as possible, by exploring the diverse putative habitats (e.g., immerged plants, aquatic vertebrates, littoral, sediments, pelagic) as well as increasing the resolution of the fungal diversity by realizing temporal surveys at the scale of single ecosystems. For example, *Wurzbacher et al. (2016)* discovered a high biodiversity of fungi and a large number of ecological niches in a single lake. They also showed that the sediment and biofilms are hotspots of aquatic fungal diversity.

Moreover, a large part of this fungal diversity seems to be active in lacustrine ecosystems (*Debroas, Hugoni & Domaizon, 2015*; *Lepère et al., 2016*). Furthermore, considering that only 1112 species of marine fungi have been described while 71% of the planet is covered by marine water (*Jones et al., 2015*), it is very likely that the global fungal diversity is greatly underestimated in aquatic systems.

### Funding
This work was partly supported by the SENDEFO project funded by the French ANR (National Research Agency) Contaminants- Ecosystems-Health (CES-2009). The funders had no role in study design, data collection and analysis, decision to publish, or preparation of the manuscript.

### Grant Disclosures
The following grant information was disclosed by the authors:
SENDEFO project funded by the French ANR (National Research Agency) Contaminants-Ecosystems-Health (CES-2009).

### Competing Interests
The authors declare there are no competing interests.

### Author Contributions
- Cécile Lepère performed the experiments, analyzed the data, contributed reagents/materials/analysis tools, prepared figures and/or tables, authored or reviewed drafts of the paper.
- Isabelle Domaizon, Jean-Francois Humbert and Ludwig Jardillier analyzed the data, contributed reagents/materials/analysis tools, approved the final draft.
- Mylène Hugoni performed the experiments, analyzed the data, contributed reagents/materials/analysis tools, approved the final draft.
- Didier Debroas conceived and designed the experiments, performed the experiments, analyzed the data, contributed reagents/materials/analysis tools, authored or reviewed drafts of the paper.

### Data Availability
Links to the raw data used in the meta-analysis are available in Table S2.

### Supplemental Information
Supplemental information for this article can be found online at http://dx.doi.org/10.7717/peerj.6247#supplemental-information.

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
