# Peer review of "Diversity, spatial distribution and activity of fungi in freshwater ecosystems"

_PeerJ, doi:10.7717/peerj.6247_

## Round 0.1 · original submission · Major Revisions

Please see the reviewer comments, as they have done a thorough job and I agree with them.

Reviewer 1 ·

Basic reporting

Basic reporting is fine.

Experimental design

The authors need to critically evaluate and discuss the biases associated with different DNA extraction techniques used between the different studies that they analyze. The should be a new section (e.g. at least one paragraph) added to the discussion dedicated to this important topic.

Validity of the findings

The findings are valid.

Additional comments

This manuscript deals with an interesting meta-analysis of existing fungal sequence data in public databases, and attempts to summarize the biogeography and diversity between many different studies. I thought it was highly interesting, to compare the overlap in OTUs between different freshwater and marine environments. That analysis alone will surely have a big impact. My main comments deal with a lack of information regarding why they chose to analyze the sites reported in table 1 and Figure 3, since there are alot of studies on marine systems with public sequence data available out there that were not considered (for example summarized in Richards et al. ,2012). Also, the quality of the figures as they are currently presented is rather low, and this should be improved prior to publication. Furthermore, the authors need to at least acknowledge and discuss the biases associated with the different DNA extraction techniques used between studies as this will influence which Fungi are detected or missed. Generally I think this manuscript has alot to add to the field and would recommend publication after they can address the following comments.

Comments:

lines 82-95: Since you are analyzing public data - why do you have a section on RNA extraction and amplification techniques? Or did you contribute new data here ? Please explain this very important point.

lines 145-157: what was the sequencing depth per sample? How many sequences were required to reach the plateau?

line 148: Why did you use 95% instead of 97%?

lines 172-175: It is very hard to know whether such low abundances in high throughput sequencing data are due to minute amounts of cross sample contamination during the PCR and library preparation steps.

lines 189-190: This analysis is highly interesting, and the Table S1 should be moved to the main manuscript (not hidden in the supplemental materials).

line 247: I do not believe that you can make a reliable taxonomic assignment with <80% sequence identity in the 18S V4 region. These taxa should be flagged as “unidentified”.

lines 277-278: You cannot claim species level identifications with 18S data for fungi. There is a general agreement that ITS can be used for this (see Schoch et al 2012 PNAS). Please revise this statement accordingly.

line 324: This should be presented as a main figure and not hidden in the supplemental.

lines 342-351: Please also cite Burgaud et al (2013), since this is highly relevant to the discussion on links between the terrestrial and marine fungi discussed here.

lines 353-354: Is this a study on already published data only? Reading the methods it seems there is new data being presented ? Please clarify this important point.

Figure 1: Please add error bars showing the standard deviation across the high number of sampled locations.

Figure 3 and Table 1: There are other studies that looked at the diversity of marine Fungi that are not considered. For example, Burgaud et al 2013 and Orsi et al 2013 both reported 18S sequence data for marine fungi. Do the authors have a reason why more studies were not considered? It is not clear from the text why they only chose the selected samples. I think the manuscript would benefit from a higher number of marine samples being included in this interesting analysis.

Figure 4: Please change the labeling on the x-axis to the taxonomic information associated with each OTU. Currently there is nothing to be learned from the label that is there now.

Figure 5: Please explain in the legend more exactly what the y axis is showing. Is this the percent of total DNA OTUs that were also detected in the RNA dataset?

Figure 6: Same comment as for Figure 5.

Reviewer 2 ·

Basic reporting

Many grammatical errors.
Figures 5 and 6 can be combined.

Experimental design

"Rigorous investigation performed to a high technical & ethical standard" is not met.

"Methods described with sufficient detail & information to replicate." is not met.

See my general comments for details.

Validity of the findings

"Data is robust, statistically sound, & controlled." is not supported.

Additional comments

Lepere et al. provided a neat summary of our knowledge about freshwater fungi by inventorying the 18S rRNA gene in public databases. In addition, they included a small rRNA dataset they generated themselves to evaluate the activity of freshwater fungi. While I think the topic is interesting to a broad audience, there are a few major issues with the manuscript that need to be addressed before it is adequate for publication.

First, the authors should focus their effort on freshwater fungi if they do not intend to make an effort to inventory public database for marine fungal 18S rRNA genes. Currently the authors include a few marine datasets to “compare environments and define spurious OTUs”. While the second purpose is fine, it is not appropriate to compare the complete freshwater dataset with a small fraction of the marine dataset and no meaningful conclusion can be drawn from such a comparison (e.g. lines 382-383). If the authors could provide a comprehensive analysis of the marine 18S rRNA genes in public databases, the profile and the contribution to the field of this article can be significantly enhanced. The manuscript in its current form is taking advantage of a mostly analyzed dataset from an already published study that examined all freshwater microbial eukrayotes (Debroas et al. 2017), and only serves as a “zoom-in” in part of that study.

Second, the authors included a small 18rRNA dataset they generated themselves to evaluate freshwater fungal activity, but very little detail was given on how that was done which precludes readers from interpreting the results properly.

Below are detailed comments.

Line 42: “Fungi” should be lower case.
Line 42: The authors probably want to say “little” as opposed to “a little”.
Line 48: “extend” should be “extent”.
Line 80: In what form were these public data collected? If it’s in the form of raw reads, it’s important to include in Table 1 how many reads are collected from each study.
Line 112: Specify which version of USEARCH was used for analysis.
Line 118: Justify the choice of 95% similarity threshold and discuss its implication for results.
Line 139: Citations should be provided for these two methods.
Lines 163-164: The “probably due to…” part doesn’t belong to the Results section.
Line 164: What is the definition of “reads” here? It seems unlikely that 19,008 OTUs were generated from only 1.6 million raw reads.
Lines 173-176: The implication of these statistics (Figure 2) is inconsistent with the diversity indices (Table 2). According to the diversity indices, Cryptomycota have the highest species richness. However, the OTU-to-reads ratio was actually the lowest for Cryptomycota (17.6% divided by 15%). Please explain this inconsistency.
Line 187: It should be “accounting” instead of “counting”.
Line 204: It should be “most” instead of “more”.
Lines 218-233: It is important to provide context to the sampled 18S rRNA. When were they sampled? What was the sampling procedure? None of such detail was outlined in the method section but they are necessary for interpreting the results properly. Additionally, in the discussion the authors should address whether and how the difference in sequencing platform between their own sequencing effort and previous ones may have an impact on the comparison.
Line 246: “is” should be “are”.
Lines 277-278: Why is this the case? Is this a result of the different life cycles between the chytrids and the dikarya fungi? This observation is interesting and should be looked into in greater details.
Line 288: The sentence needs to be broken down into two.
Lines 334-339: The authors seems to believe that Dikarya fungi previously identified as terrestrial should not contribute to organic matter recycling in freshwater lakes, even though their data suggest otherwise. I think the authors should either provide stronger arguments for their stand, or not jump to the conclusion since there is much conflicting literature on this topic.
Line 362: Why is this FISH image buried in the supplemental material? Is it published somewhere else? I think the authors should move it to the main text, and describe the associated methods.
Line 364: “term” should be “terms”.
Lines 382-383: This statement is unwarranted because the authors did not inventory the marine fungi 18S rRNA gene database.
Figure 5: Why are Basidiomycota and Ascomycota lumped as Dikarya here while they are not in the other figures? Basidiomycota and Ascomycota should be shown as separated groups.
Figure 5 and Figure 6 can easily be integrated into one figure and be more informative.

---

## Round 0.2 · accepted · Accept

Reviewer 2 was unable to re-review the article, but I can confirm that their concerns were appropriately addressed. Please note the comments from Reviewer 1 about including all references. Thanks!

# Reviewer 1 ·

Basic reporting

no comment

Experimental design

no comment

Validity of the findings

no comment

Additional comments

The authors have done a good job responding to my comments and I now recommend this manuscript for publication. Please note however, that some of the newly added citations are missing from the reference list.